# No evidence of sustained nonzoonotic *Plasmodium knowlesi* transmission in Malaysia from modelling malaria case data

Kimberly M. Fornace ●[1,2,3] ✉, Hillary M. Topazian[4], Isobel Routledge[4,5], Syafie Asyraf[6], Jenarun Jelip[7], Kim A. Lindblade ●[8], Mohammad Saffree Jeffree ●[6], Pablo Ruiz Cuenca[3], Samir Bhatt[4,9], Kamruddin Ahmed ●[6], Azra C. Ghani ●[4] & Chris Drakeley ●[3]

Reported incidence of the zoonotic malaria *Plasmodium knowlesi* has markedly increased across Southeast Asia and threatens malaria elimination. Non-zoonotic transmission of *P. knowlesi* has been experimentally demonstrated, but it remains unknown whether nonzoonotic transmission is contributing to increases in *P. knowlesi* cases. Here, we adapt model-based inference methods to estimate $R_C$, individual case reproductive numbers, for *P. knowlesi*, *P. falciparum* and *P. vivax* human cases in Malaysia from 2012–2020 (n = 32,635). Best fitting models for *P. knowlesi* showed subcritical transmission ($R_C < 1$) consistent with a large reservoir of unobserved infection sources, indicating *P. knowlesi* remains a primarily zoonotic infection. In contrast, sustained transmission ($R_C > 1$) was estimated historically for *P. falciparum* and *P. vivax*, with declines in $R_C$ estimates observed over time consistent with local elimination. Together, this suggests sustained nonzoonotic *P. knowlesi* transmission is highly unlikely and that new approaches are urgently needed to control spillover risks.

*Plasmodium knowlesi* is a zoonotic malaria parasite carried by simian reservoirs and transmitted to people through bites of infected *Anopheles* mosquitoes. Since the first report of hundreds of human *P. knowlesi* cases in Malaysian Borneo in 2004, reported *P. knowlesi* incidence has markedly increased across Southeast Asia[1,2]. Although increases in cases are partially due to improved detection with the application of molecular diagnostics, data indicate genuine increases in both the proportion and incidence of *P. knowlesi*[3,4]. In Malaysia, no indigenous cases of nonzoonotic malaria have been reported since 2018 and *P. knowlesi* is now the main cause of malaria in humans and can cause severe and fatal malaria when untreated[5,6]. These increases in

reported incidence are strongly associated with deforestation, suggesting that landscape and land use change may be increasing contact between people, mosquitoes and macaque reservoirs of *P. knowlesi*[7,8]. *P. knowlesi* risk in humans is highly spatially heterogenous, with clusters of cases reported in households and villages[9–11]. This may reflect similar spatial variation in the force of infection from wildlife reservoir populations or increased adaptation to transmission between human hosts.

Many emerging pathogens have both zoonotic and nonzoonotic transmission pathways in which an initial spillover event from an animal reservoir may lead to stuttering chains of human-to-human transmission or larger outbreaks[12]. For *P. knowlesi*, genetic evidence suggests that

[1]School of Biodiversity, One Health and Veterinary Medicine, University of Glasgow, Glasgow, UK. [2]Saw Swee Hock School of Public Health, National University of, Singapore, Singapore. [3]Faculty of Infectious and Tropical Diseases, London School of Hygiene and Tropical Medicine, London, UK. [4]MRC Centre for Global Infectious Disease Analysis, Imperial College London, London, UK. [5]University of California, San Francisco, San Francisco, USA. [6]Faculty of Medicine and Health Sciences, Universiti Malaysia Sabah, Kota Kinabalu, Malaysia. [7]Vector-borne Disease Control Division, Ministry of Health Malaysia, Putrajaya, Malaysia. [8]Global Malaria Programme, World Health Organization, Geneva, Switzerland. [9]Section of Epidemiology, University of Copenhagen, Copenhagen, Denmark. ✉e-mail: Kimberly.Fornace@lshtm.ac.uk

transmission to humans remains primarily zoonotic, occurring through bites from vectors infected by wild non-human primate populations[13]. Evidence suggests *P. knowlesi* infections are typically chronic with low-level parasitemias in long-tailed macaques (*Macaca fascicularis*), the main reservoir[14]. The spatial distribution of human cases mirrors *P. knowlesi* infection rates in macaques, with the highest infection rates in humans and macaques reported in Malaysian Borneo[15]. *P. knowlesi* has a rapid 24 h replication rate with observed prepatent periods in humans ranging from 9 to 12 days[16]. Although limited data are available on the duration and frequency of human infectiousness to mosquitoes, the feasibility of human–mosquito–human transmission of *P. knowlesi* has been demonstrated once experimentally in a laboratory study[16] but has not been formally described in natural settings. A recent systematic literature review concluded nonzoonotic *P. knowlesi* transmission was biologically plausible, but there was no empirical evidence that human–mosquito-human *P. knowlesi transmission* occurs naturally[17]. Notably, no previous studies had systematically assessed whether spatiotemporal patterns of *P. knowlesi* cases were consistent with nonzoonotic transmission.

Quantifying the relative contributions of zoonotic and nonzoonotic transmission is essential for designing effective surveillance and control. Transmission pathways are represented by two epidemiological parameters: the spillover rate, the rate at which a pathogen is transmitted from animals to humans, and the reproductive number ($R$), the number of secondary human cases resulting from an infectious human individual over the course of their infection[18]. Directly attributing human infections to zoonotic sources is challenging as the distributions and prevalence of wildlife reservoirs are largely unknown, and infection occurs indirectly via mosquito vectors. Alternatively, model-based inference methods can be used to infer likely transmission routes from routinely collected surveillance data on reported cases[18–21]. Diffusion network approaches have been used to quantify individual malaria case reproduction numbers ($R_C$) by evaluating the probability of two cases being connected using the time of symptom onset, spatial locations of cases, and estimated serial interval (SI), i.e. the time from symptom onset in one case to symptom onset in the secondary case[19,22–24]. These methods also allow assessment of the likelihood of cases occurring from outside (unobserved) sources not represented in human surveillance data, such as introductions through zoonotic spillover[18,21].

Here, we use a diffusion network approach within a Bayesian framework to estimate $R_C$ for *P. knowlesi* and nonzoonotic malaria species across Malaysia from 2012 to 2020. Due to an active malaria elimination programme, Malaysia has an exceptionally strong surveillance system, and detailed information is available for all cases. We use this surveillance dataset and parameters identified from the literature to estimate the SI for nonzoonotic *P. knowlesi* transmission. We then quantify estimates and uncertainty around $R_C$ for the zoonotic malaria

*P. knowlesi* and nonzoonotic malaria *P. falciparum* and *P. vivax*. Models of *P. knowlesi* transmission were consistent with a large reservoir of unobserved infection sources, suggesting human cases are primarily driven by spillover and with limited subcritical onward transmission ($R_C < 1$). In contrast, $R_C$ estimates for *P. falciparum* and *P. vivax* identified historical sustained human–mosquito–human transmission (i.e. $R_C > 1$), with subsequent decreases in $R_C$ observed over time as the country approached elimination of transmission of these two nonzoonotic malaria parasites.

## Results

Between 16 December 2011 and 3 January 2021, 32,635 malaria cases were reported to the Malaysian national malaria surveillance system. Of these, the majority ($n = 26,093$, 79.95%) were reported from the East Malaysian states of Sabah and Sarawak (Fig. 1). All suspected *P. knowlesi* cases were confirmed by molecular methods, with confirmed *P. knowlesi* accounting for over 70% ($n = 23,143$) of malaria cases nationally and no indigenous nonzoonotic malaria cases reported after 2018 (Fig. 2). Based on travel history to malaria-endemic areas within the past month and previous malaria diagnoses, cases are classified by the Malaysian Ministry of Health as indigenous, introduced, imported or reoccurrences (for *P. vivax*). Most malaria cases were classified as indigenous ($n = 27,125/32,635$, 83.12%), including over 99% of *P. knowlesi* cases. Over 80% ($n = 18,691/23,143$) of *P. knowlesi* cases were reported in men, with a median age of 38 years (interquartile range [IQR]: 26–49 years). Similarly, most nonzoonotic malaria cases were also reported in adult men; the median age was 31 (IQR: 21–43 years), and 84% (7945/9489) were male.

Malaysia has high health system coverage with free treatment for malaria[25]. For available surveillance data, the median time between symptom onset and treatment was 5 days, with over 97% of individuals seeking treatment within 2 weeks (Fig. 3a). Out of 16,765 *P. knowlesi* cases screened for gametocytes by microscopy, 5.74% (95% confidence interval [CI]: 5.39–6.10%) had microscopically detectable gametocytes, consistent with other reports on the *P. knowlesi* gametocyte carriage in clinical cases[26]. Infectiousness to mosquitoes is primarily dependent on the density of malaria gametocytes, with infectious individuals predominantly having gametocyte densities high enough to be detected microscopically[27,28]. Proportions of infectious *P. knowlesi* cases increased with time since symptom onset; however, due to the relatively rapid reporting and treatment times, the mean duration of human infectiousness was only 1.645 days (range: 0–27 days) (Fig. 3b, Supplementary Information).

We assessed the likelihood of cases being part of the same transmission chain based on species-specific SI parameters, dates and geographical locations of reported cases, and case classification as indigenous or non-indigenous (imported, introduced or

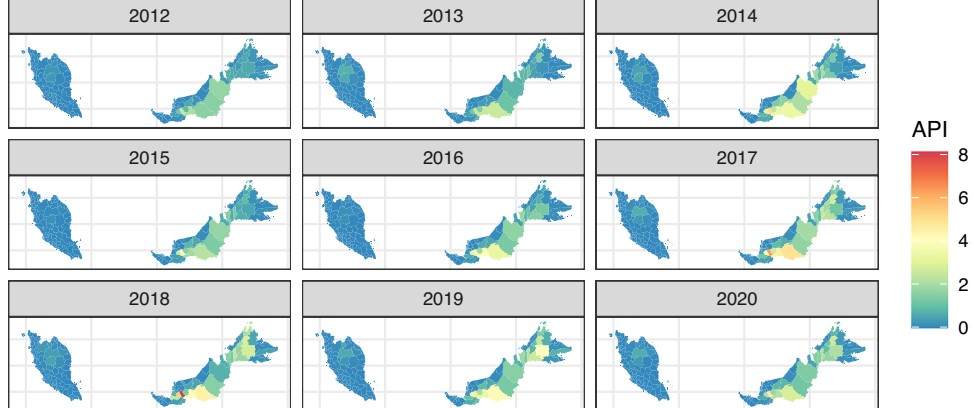

**Fig. 1 | Annual *P. knowlesi* incidence in Malaysia from 2012–2020.** Annual parasite incidence (API) of *P. knowlesi* per district, API is presented as the annual number of malaria cases per 1000 people.

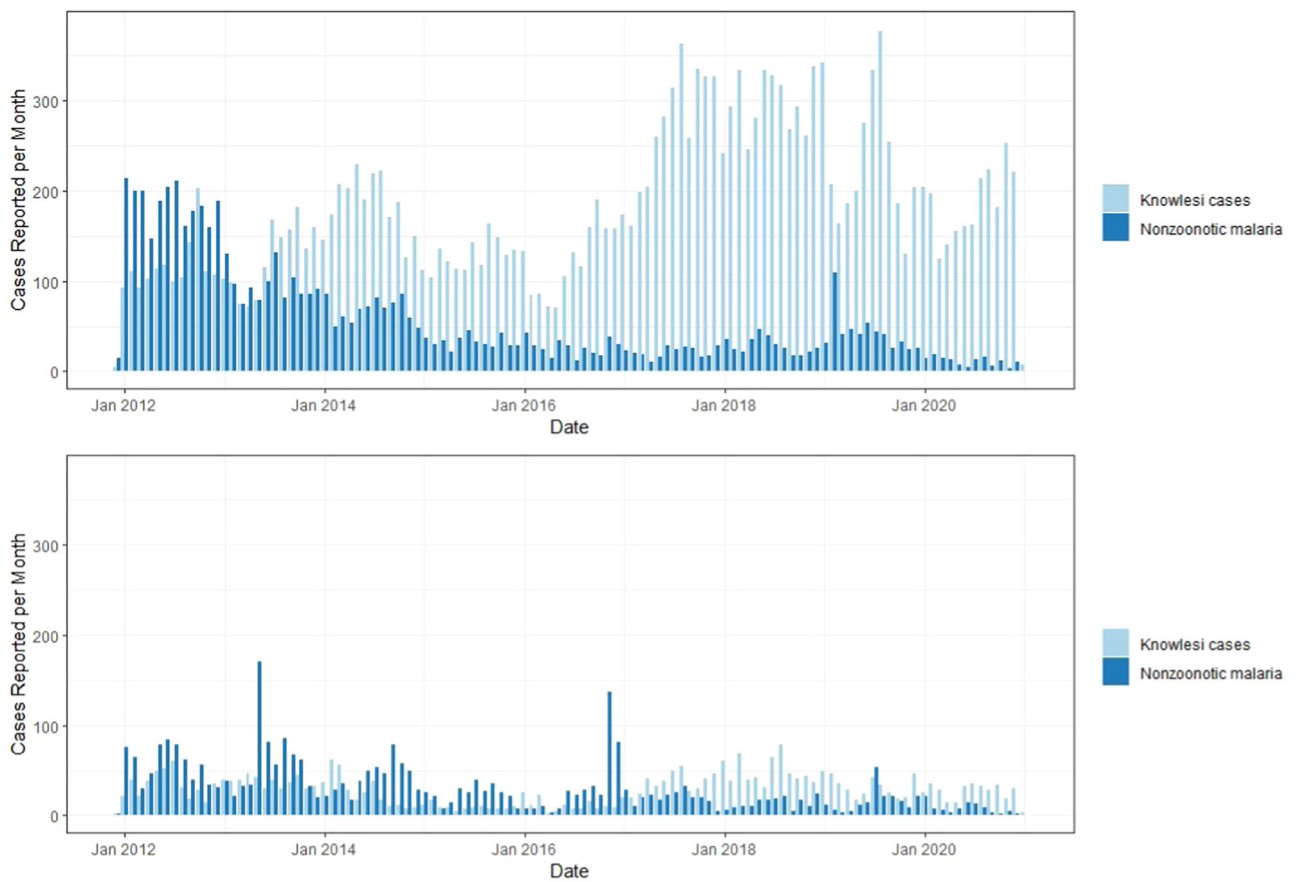

**Fig. 2 | Monthy numbers of reported *P. knowlesi* cases and nonzoonotic malaria cases.** Plots show the total numbers of reported cases for *P. knowlesi* and for nonzoonotic malaria species (*P. falciparum, P. vivax, P. malariae, P. ovale*) for East Malaysia (Sabah and Sarawak, top plot) and West Malaysia (bottom).

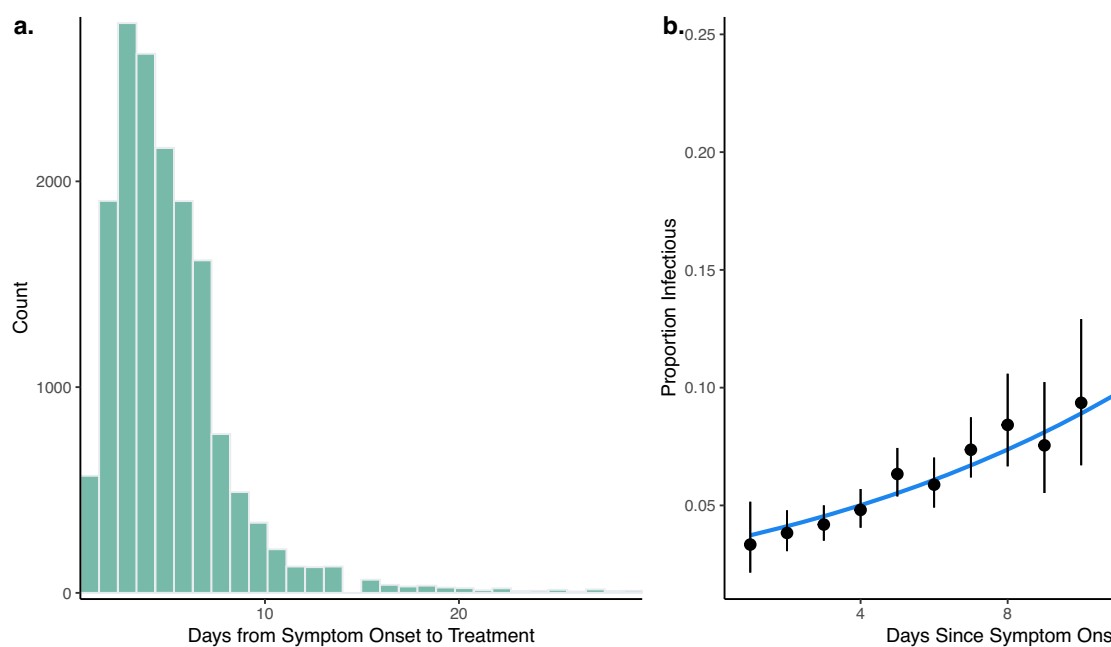

**Fig. 3 | Duration of infectiousness from surveillance data.** The time of infectiousness was calculated from **a** time between symptom onset and treatment for *P. knowlesi* cases and **b** proportion and 95% CI of clinical *P. knowlesi* cases having microscopically observed gametocytes on presentation to health facility. Data are presented as the observed proportion of cases with microscopically observed gametocytes from 16,765 clinical *P. knowlesi* cases with the error bars representing 95% confidence intervals, the line is predicted probability; see Supplementary Fig. 3.

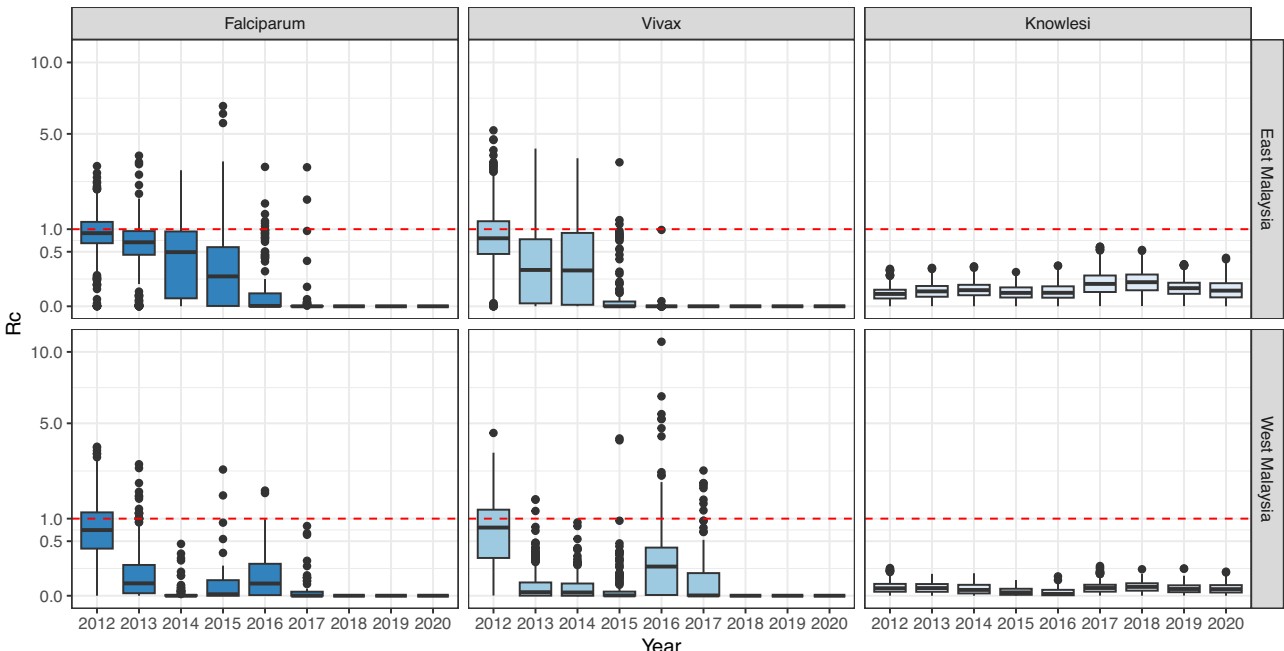

**Fig. 4 | Case reproductive numbers for *P. falciparum*, *P. vivax* and *P. knowlesi*.** $R_C$ estimates per year for *P. falciparum*, *P. vivax* and *P. knowlesi* in East and West Malaysia. Data are presented as the median, first and third quartiles and range, with points more than 1.5 times the interquartile range plotted separately as outliers.

reoccurrences). This modelling framework additionally included a threshold parameter ($\varepsilon$) that prevents the likelihood from becoming zero due to a complete break in the transmission chain and, therefore, epidemiologically represents the probability of infection from unobserved external sources needed to sustain transmission. We assessed $\varepsilon$ ranges using mean prior values ranging from <0.001, a prior distribution centring around the assumption that nearly all cases are infected from human sources included in the surveillance data to 1, a prior distribution assuming that all human cases are infected from external zoonotic or imported sources (Supplementary Table 4)[19]. *P. knowlesi* models with $\varepsilon$ values approaching 1 resulted in better fits, implying that all or nearly all human cases are infected from external sources, most likely to be zoonotic spillover from non-human primates in this context (Supplementary Tables 4 and 5). For each individual, their estimated number of secondary infections was calculated to give a time-varying point estimate of $R_C$. An individual $R_C$ value < 1 indicates a low likelihood that the case is the source infection for any other case in the dataset. Mean $R_C$ values for *P. knowlesi* in East Malaysia from 2012 to 2020 were 0.0739 (95% CI: 0.0006–0.278) and 0.0147 (95% CI: 0.000003–0.0637) for West Malaysia. There were no estimates of $R_C$ greater than 1 in the best-fitting models, and no scenarios assessed that had mean $R_C$ values greater than 1, indicating no sustained transmission (Supplementary Information). As we estimated time-varying $R_C$ values for each individual, the estimates of all $R_C$ values as less than 1 also suggest there is no major overdispersion, excluding large super-spreading events. Although there were clear increasing trends in the numbers of reported *P. knowlesi* cases through the study period, this was not reflected in increases in $R_C$ estimates, and there was no evidence of deviation from zoonotic–mosquito–human transmission patterns.

In contrast, models for nonzoonotic malaria identified evidence for clear indigenous human-mosquito-human transmission chains, with 17% of *P. falciparum* cases and 24% of *P. vivax* cases with estimated $R_C$ values greater than 1 from 2012 to 2020. For *P. falciparum*, the mean $R_C$ value was 0.568 (95% CI: 0–1.97) for East Malaysia and 0.386 (95% CI: 0–2.04) for West Malaysia. The mean $R_C$ for *P. vivax* was 0.524 (95% CI: 0–2.436) and 0.230 (95%CI: 0, 1.820) for East and West Malaysia, respectively (Fig. 4). The probability of *P. falciparum*

or *P. vivax* cases leading to onward transmission varied over space and time (Fig. 5). There were clear decreasing temporal trends in $R_C$ estimates for both nonzoonotic malaria species, with no $R_C$ values above 1 detected after 2018 when the last indigenous nonzoonotic malaria case was reported by the Malaysian Ministry of Health. Models clearly identified an outbreak of *P. vivax* reported in West Malaysia between 2016 and 2017, demonstrating the utility of these methods in pre-elimination settings[29].

## Discussion

Quantifying rates of nonzoonotic transmission is critical to understanding the dynamics of emerging zoonotic diseases and identifying appropriate interventions. While many vector-borne and zoonotic diseases have emerged to become widespread in human populations (e.g. *P. knowlesi*, Lyme disease), the increase in human case numbers is not necessarily correlated with increased nonzoonotic transmission[30,31]. Here, we demonstrate the use of quantitative model-based approaches to identify key parameters and estimate individual reproductive numbers of the zoonotic and vector-borne disease *P. knowlesi*. Model results strongly suggest that *P. knowlesi* remains a primarily zoonotic disease driven by spillover, finding no evidence of sustained nonzoonotic *P. knowlesi* transmission. These results sharply contrast with $R_C$ estimates of the nonzoonotic malaria *P. falciparum* and *P. vivax*, where clear chains of human-mosquito–human transmission were identified. This highlights the utility of this method to assess the probability of nonzoonotic transmission using high-quality routine surveillance data and provides a critical tool to monitor changing transmission patterns.

A key determinant of the probability of nonzoonotic *P. knowlesi* transmission is whether humans are infectious. Using the largest dataset of gametocyte carriage rates in *P. knowlesi* clinical cases reported, consistent proportions of cases had microscopically observed gametocytes (5.74%), supporting the possibility of non-zoonotic *P. knowlesi* transmission. However, *P. knowlesi* gametocyte carriage rates in clinical cases upon presentation to health facilities are notably lower than for other nonzoonotic malaria species. For example, for *P. vivax* infections, which are biologically similar to *P. knowlesi*, up to 100% of clinical cases have microscopically observed

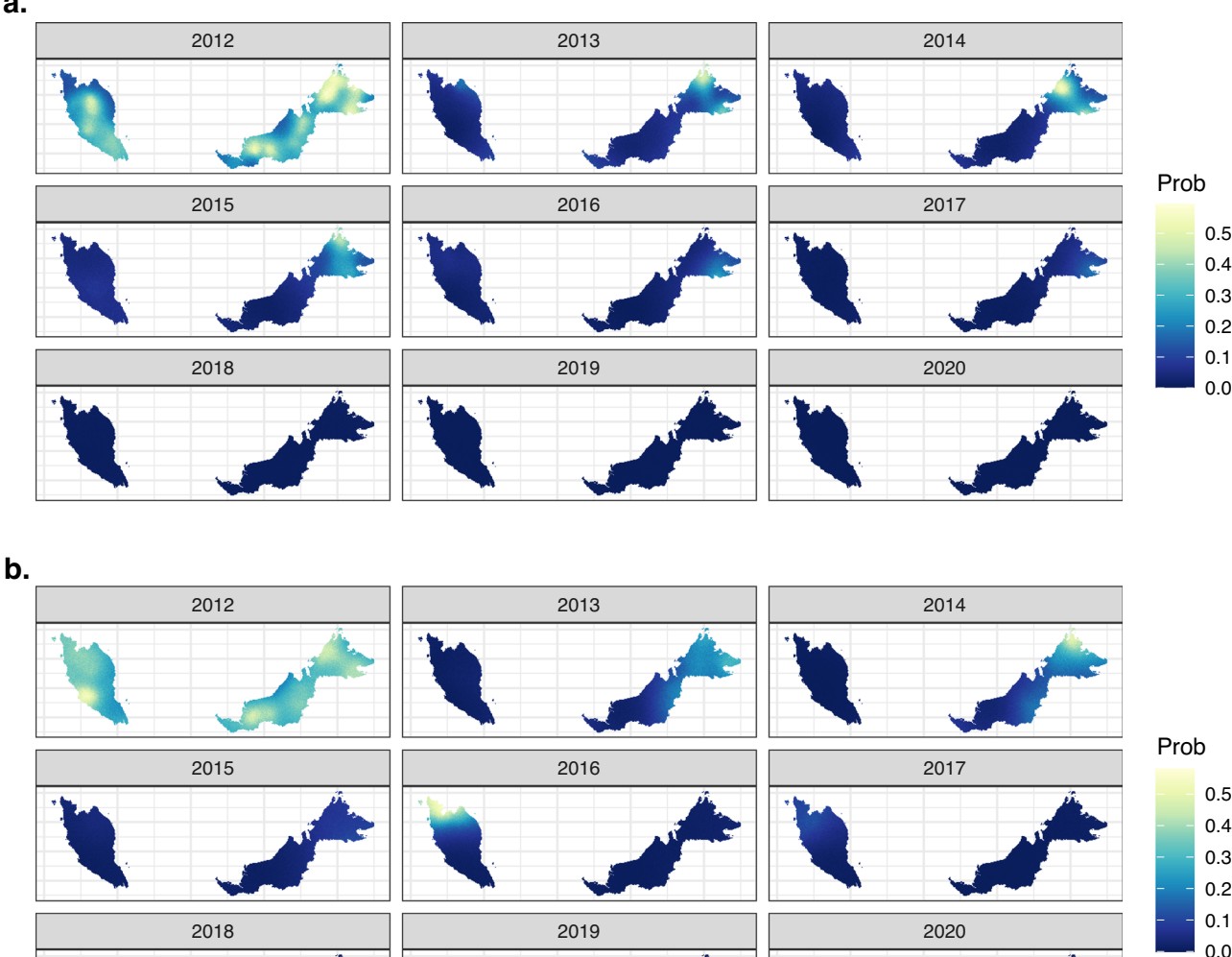

**Fig. 5 | Mean posterior probability of $R_C$ (case reproductive number) estimates exceeding 1.** The probability of $R_C$ exceeding 1 is shown for **a** *P. falciparum* and **b** *P. vivax*; *P. knowlesi* probabilities were not calculated as there were no estimates of $R_C$ greater than 1 in the best-fitting models, but village-level $R_C$ estimates can be found in Supplementary Fig. 3.

gametocytes at presentation[32]. More extensive meta-analyses of gametocyte carriage in *P. falciparum* cases have also identified a wider range (12.1–32.2%) of microscopically observed gametocytes at rates typically higher than *P. knowlesi* cases reported within this dataset[33]. While this does suggest human *P. knowlesi* cases have the potential to be infectious, this parasite may be less adapted to human hosts in terms of gametocytogenesis. Critical questions remain about how infective *P. knowlesi*-infected humans are to mosquitoes, particularly considering the strong circadian rhythms of infectiousness to mosquitoes observed in *P. knowlesi*-infected non-human primates[34].

Our model fits strongly suggests that *P. knowlesi* remains primarily driven by zoonotic spillover, finding no evidence of sustained nonzoonotic *P. knowlesi* transmission. For all of the best fitting models, there were no estimates of $R_C$ greater than 1, and in all scenarios assessed, there were no estimates suggesting $R_C$ is routinely over 1. There was additionally no clear evidence of changes to *P. knowlesi* $R_C$ over time that would suggest *P. knowlesi* may be adapting to transmission between humans. This contrasts with $R_C$ estimates for both *P. falciparum* and *P. vivax*, where $R_C$ estimates frequently exceed 1 and show clear decreasing temporal trends during an extensive malaria elimination campaign. This highlights the utility of these methods in monitoring changing transmission patterns of both emerging and endemic vector-borne diseases[19,20].

While these results can shed insight into broad transmission patterns, this analysis is not without significant limitations. Model-based estimates alone cannot be used to conclusively prove that two cases are linked. This modelling approach estimated $R_C$ under different assumptions but did not reconstruct transmission networks linking cases. Further genetic, epidemiological and ecological data would be needed to prove transmission occurred between two individuals and rule out multiple spillover events occurring from zoonotic reservoirs. Additionally, while *P. knowlesi* results are consistent with a large reservoir of unobserved infections, this modelling framework does not explicitly attribute these to zoonotic sources. Other potential sources include asymptomatic or unreported infections. However, given the high coverage of the Malaysian surveillance system, infrequency and low parasite density of asymptomatic human *P. knowlesi* infections[35,36] and widespread distribution of infected simian reservoirs[15], unobserved infections would appear to be mostly from zoonotic sources. As higher rates of low-density, asymptomatic *P. knowlesi* infections have been reported in people residing near clinical cases, further studies are needed to identify whether peri-domestic or localised nonzoonotic *P.*

*knowlesi* transmission is occurring but not detected by clinical surveillance systems[35]. Higher estimates of $R_C$ values could be used to guide targeted investigations at locations with the highest likelihood of nonzoonotic transmission. Future studies could additionally explore geographic variability in human–mosquito contact rates, human immunity, treatment-seeking behaviours and other factors impacting transmission probabilities and SI distributions.

Despite these limitations, results indicate there is no evidence of sustained nonzoonotic *P. knowlesi* transmission and demonstrate decreases in the transmission of *P. falciparum* and *P. vivax* in Malaysia throughout the past decade. This has important implications for the design of malaria surveillance and control programmes in areas with zoonotic malaria. Current measures, such as insecticide-treated net distribution and active case detection, have proven highly effective at controlling nonzoonotic malaria. While these measures may also be reducing nonzoonotic *P. knowlesi* transmission, new control strategies are urgently needed to mitigate the risks of spillover from wildlife hosts. With the inclusion of zoonotic malaria in global malaria elimination certification criteria, *P. knowlesi* poses a major threat to malaria elimination across Southeast Asia[37]. These methods provide a template for monitoring potential changes in transmission patterns of *P. knowlesi* and other zoonotic and vector-borne diseases.

## Methods

### Ethics
Ethical approval for this study was granted by the Medical Ethics and Research Committee of the Ministry of Health Malaysia (NMRR-21-470-58703) and the London School of Hygiene and Tropical Medicine Research Ethics Committee (Ref. 25654). The study protocol was registered under the National Medical Research Registry, Malaysia.

### Surveillance data
National malaria surveillance data collected by the Malaysian Ministry of Health were assembled from 16 December 2011 to 3 January 2021. Data on malaria cases included demographic data (age, gender and occupation), case notification date to the Ministry of Health, symptom onset date, diagnosis date, method of detection, importation status, parasite species diagnosed, presence of gametocytes and health facility of detection. Symptom onset dates were excluded as unreliable if they occurred after diagnosis or notification dates or more than 30 days prior to diagnosis. Due to frequent misdiagnosis of *P. knowlesi* by microscopy, diagnosis of all presumptive *P. knowlesi* cases and imported malaria, cases were confirmed using species-specific molecular methods[25]. Additionally, surveillance data included geographical data on the case residence, probable location of the infection (state, district, village name and/or GPS coordinates) and travel history. All case locations were manually confirmed and geolocated to the nearest village centroid (Supplementary Information). Cases without accurate address data were excluded from the final analysis ($n = 2117$).

### SI estimation
The SI, the time between symptom onset of primary and secondary malaria cases, requires estimating the duration of a series of sequential processes which need to occur in a human–mosquito–human transmission cycle[38]. While these intervals can be inferred from contract tracing data for directly transmitted diseases, SIs can only be estimated indirectly for vector-borne diseases[39,40]. SI durations have previously been estimated for *P. falciparum* and *P. vivax*[19,40]. However, there is a lack of empirical evidence on human–mosquito–human *P. knowlesi* transmission as this has only been observed once under experimental conditions[16]. The rapid replication cycle of *P. knowlesi* and weak evidence of adaptation to humans suggests the SI may differ from that of other nonzoonotic malaria parasites[34].

To estimate the SI for human–mosquito–human *P. knowlesi* transmission, we adapted a quantitative model-based approach[40]. This models the SI as the sum of random variables representing sequential steps in the transmission cycle, including the prepatent period, human to mosquito transmission period, extrinsic incubation period, mosquito-to-human transmission period and infection-to-reporting periods. These were parameterised based on a combination of data from secondary literature and the Malaysian malaria surveillance dataset (Supplementary Information). Due to the typically low parasite densities and infrequency of asymptomatic human *P. knowlesi* infections in the region, we assumed asymptomatic infections did not contribute to transmission[35,41]. We also assumed individuals were not infectious after reporting due to Malaysia's mandatory treatment and hospitalisation policy. Finally, no data were available on the duration or timing of infectiousness of *P. knowlesi* in humans, and the relatively short duration between symptoms onset and treatment within this population precluded estimation of natural recovery rates. While *P. knowlesi* gametocytes are highly synchronous and have strong circadian rhythms in macaques[34,42], available evidence suggests this is not the case in human infections[43].

### Transmission likelihood and $R_C$ estimation
To estimate the joint likelihood of transmission between cases and a case having an unobserved source of infection, we used an adapted version of the NetRate algorithm to calculate $R_C$ for each case[19,20,22,44]. This uses a Bayesian framework to estimate the likelihood of two cases being within the transmission chain based on the geographical locations of cases, indigenous case status, time of symptom onset and SI parameters (Supplementary Information). Case reproduction numbers ($R_C$) are then estimated by, for each case, summing the normalised likelihood of that case being the infector of each of the other cases. Non-indigenous cases and cases with a symptom onset date before the candidate case are not included as potential infectees. This approach additionally includes a provision for missing or unobserved sources of infections that can infect any observed individual with probability $\varepsilon$. These epsilon edges ($\varepsilon$) link cases to an external source when the likelihood of transmission between two cases is sufficiently low. Higher values of $\varepsilon$ indicate a larger proportion of cases are assumed to be infected by external sources outside the surveillance data; in the case of *P. knowlesi*, this is mostly to be zoonotic reservoirs.

We fit separate models for *P. knowlesi*, *P. falciparum* and *P. vivax* for both East Malaysia (Borneo) and West Malaysia. From surveillance reports, most cases reported only local travel within the same village or subdistrict. To account for a high likelihood of local transmission, we used a fixed value for $\delta$ of 0.1, the parameter determining transmission likelihood based on Euclidean distance between cases (assessed in kilometres); this corresponds to most transmission events occurring within 10 km based on reported travel history (Supplementary Information). We fit shifted Rayleigh distributions to describe a prior distribution of possible SIs for *P. knowlesi* and nonzoonotic malaria. This distribution is defined by $\alpha$, the shape of the SI distribution, and $\gamma$, the shifting parameter accounting for the minimum time between a bite from an infected mosquito and infectiousness in a human. For *P. falciparum* and *P. vivax*, we used previously published parameters to inform the SI distribution, with $\gamma$ fixed at 15 days and normally distributed priors on $\alpha$ with a mean of 0.003 and a standard deviation of 0.1; this corresponds to a mean SI of 36 days (95% range: 21–60 days)[20]. For *P. knowlesi*, the $\gamma$ was fixed at 14 days based on the modelled time to infectiousness fit from the Malaysian surveillance data (Supplementary Information). To account for uncertainty in the SI distribution, we used a normally distributed prior for $\alpha$ with a mean of 0.002 and standard deviation of 0.001, capturing the full range of plausible estimates of the *P. knowlesi* SI.

For each malaria species and location, surveillance data was input as a time-ordered series of malaria notification dates. As there was

considerable uncertainty in the proportion of zoonotic transmission of *P. knowlesi*, we conducted a sensitivity analysis using normally distributed priors for $\varepsilon$ with mean values of 0.01, 0.1 and 1. As $\varepsilon$ values were bounded between 1e$^{-11}$ and 1, prior estimates were truncated at 1, forming a half-normal estimate for higher values. Similar sensitivity analysis was conducted for *P. falciparum* and *P. vivax*, with mean values of priors for $\varepsilon$ ranging from 0.0001 to 0.000001, reflecting the expected high sensitivity of malaria surveillance systems. We assessed model fit based on second-order Akaike's Information Criteria (AIC) and visually inspected the posterior distributions of key parameters ($\gamma$, $\delta$, $\varepsilon$) to select the best-fitting model. For all final models, we estimated $R_C$ by creating a matrix of potential infectors and infectees with the normalised likelihood of cases being connected. For each individual, the fractional number of secondary infections was calculated to give a time-varying point estimate of $R_C$. Models were previously validated through simulation studies[44], with code available at https://github.com/IzzyRou/spatial_rcs.

Spatial and temporal patterns of $R_C$ were explored for all species and locations. For *P. falciparum* and *P. vivax*, the probability of onward transmission ($R_C > 1$) was assessed using statistical models with joint temporally and spatially structured random effects with Bayesian inference approximated using the Integrated Nested Laplace Approximation (INLA)[45]. We classified cases into dichotomous categories based on $R_C$ estimates. For each species, we fit a binomial model with a logit link to the number of cases with $R_C$ estimates > 1 out of the total cases per year detected within 5 km$^2$ grid cells (Supplementary Information). The best-fitting model was selected using Deviance Information Criteria (DIC) and included a spatial effect and a temporally structured random walk model of order 2[46]. The spatial effect was modelled as a Matern covariance function using an approximation to a stochastic partial differential equations approach. Posterior expectations and probabilities were estimated using 1000 samples. All data analysis was conducted in R statistical software v 4.1, and data were checked and cleaned using Quantum GIS v 3.24.

### Reporting summary

Further information on research design is available in the Nature Portfolio Reporting Summary linked to this article.

## Data availability

As malaria case surveillance datasets contain identifiable health information, data is only available following approval of the Medical Ethics and Research Committee in Malaysia, relevant ethics committees in the UK and with permission from the Malaysian Ministry of Health. Requests to access datasets should be directed to Kimberly.Fornace@lshtm.ac.uk.

## Code availability

Code and simulated datasets are available at https://github.com/IzzyRou/spatial_rcs.

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

## Acknowledgements

This work was funded by the World Health Organisation (WHO). KMF is supported by a Sir Henry Dale Fellowship jointly funded by the Wellcome Trust and the Royal Society (Grant No. 221963/Z/20/Z). S.B., A.G. and H.M.T. acknowledge support from the MRC Centre for Global Infectious Disease Analysis (MR/R015600/1), jointly funded by the UK Medical Research Council (MRC) and the UK Foreign, Commonwealth & Development Office (FCDO), under the MRC/ FCDO Concordat agreement, and also part of the EDCTP2 programme supported by the European Union. HMT and AG are supported by the Wellcome Trust (Grant No. 220900/Z/20/Z). S.B. acknowledges support from the Novo Nordisk Foundation via The Novo Nordisk Young Investigator Award (NNF20OC0059309) and the Danish National Research Foundation via a chair position. S.B. acknowledges support from The Eric and Wendy Schmidt Fund for Strategic Innovation via the Schmidt Polymath Award (G-22-63345). S.B. acknowledges support from the National Institute for Health Research (NIHR) via the Health Protection Research Unit in Modelling and Health Economics. The authors would like to thank the Director General of the Ministry of Health in Malaysia for giving permission to publish this paper. For the purpose of open access, the author has applied a "Creative Commons Attribution (CC BY) licence (where permitted by UKRI, "Open Government Licence" or "Creative Commons Attribution No-derivatives (CC-BY-ND) licence may be stated instead) to any Author Accepted Manuscript version arising.

## Author contributions

Conceived study: K.M.F., J.J., K.L., S.B., M.S.J., K.A., A.G. and C.D.; collected and collated data: J.J. (Malaysian Ministry of Health), K.M.F., S.A., P.R.C. and K.A.; carried out analysis: K.M.F., H.M.T., I.R., S.A., S.B., A.G. and C.D.; wrote paper: K.F.; commented and reviewed drafts: all authors.

## Competing interests

Dr. Kim Lindblade was a staff member of the World Health Organisation at the time of this work. The authors alone are responsible for the views expressed in this article, and they do not necessarily represent the decisions, policies or views of the World Health Organisation. The remaining authors declare no competing interests.
