## [Peer Review File · Nature Communications]

No evidence of sustained nonzoonotic *Plasmodium knowlesi* transmission in Malaysia from modelling malaria case dataREVIEWER COMMENTS

Reviewer #1 (Remarks to the Author):

The study explored the possibility of nonzoonotic transmission of *P.knowlesi* by fitting a mathematical model to cases reported in Malaysia. The model attempted to quantify the individual malaria case reproduction number R_c , which is the number of secondary human-mosquito-human cases from an index case, informing by three main quantities 1) the time of symptom onset, 2) spatial location of cases, and estimated serial interval (SI). The results shown that the R_c estimate of *P.knowlesi* is less than one, supporting that the feasibility of human-mosquito-human transmission is low. The authors also applied this same model to cases reported for *P.falciparum* and *P.vivax* and the results shown their R_c are larger than one in the past, confirming the historical human-to-human outbreaks of these malaria species. The findings of this manuscript is interesting, as nonzoonotic transmission of *P.knowlesi* was previously confirmed to be feasible in vivo but no evidence was found in epidemiological data. Defining the exact mode of transmission will help to optimize resources to prevent transmission of this malaria species. The methods used here was sound and interesting and could be used for other zoonosis diseases with available case report data. The manuscript was well written, clear, and ready to publish but still can be benefit from some edits and suggestions.

Since the methods had a section focused on estimating SI, I think the introduction should briefly include some information about the natural progression of a *P.knowlesi* case, like incubation period, infectious period, recovery rate, proportion of mortality or disability. This information was scattered around the manuscript but without a systematic summary beforehand, it may confuse people who is not familiar with the disease.

Related to the calculation of SI, I find it confusing that the proportion of infectious increases with the days since symptom onset. I wonder if this is the natural progression of the disease since I expect the proportion of infectious will eventually go down as people recover. The supplement information mentioned that the proportion infectious was kept to be the same after day 14, this information should be mentioned in the main text.

One of the critics in the model when I was reading the main text is that the geographic variation of Malaysia was not accounted for when deriving the SI, but I did eventually see this point made in the supplement information. I think it is worth to bring this issue up to the main text and discuss it as a limitation of the study.

Lines 240-244: this point could be clearer if the authors articulate how the results going to change if they did not account for the asymptomatic or unreported infections. The authors should also include some information about the competency of asymptomatic transmission.

Although the authors have documented the details of their methods in the supplement information, codes should still be required for the ease of reproducing the results.

Reviewer #2 (Remarks to the Author):

ASSESSMENT

In this study, Fornace et al. leveraged high-resolution data from Malaysia to address whether there is sustained human-mosquito-human transmission of *Plasmodium knowlesi*. The authors use a well-established transmission network approach to estimate R_c , the basic reproduction number under control. They conclude that, because the estimated R_c falls below one and lies very close to zero, there is no evidence of sustained human-mosquito-human transmission of *P. knowlesi*. I really enjoyed this study and find the conclusions compelling. I mainly just have clarification and recommendations that I think will strengthen the paper when it is published in its final form. In particular, I believe that this study would benefit from a simulation study to prove that the inference framework is capable of estimate the correct values of R_c as well as sensitivity analyses

to evaluate the sensitivity of the conclusions to the choice of ϵ and δ . Finally, I found the inclusion of the *P. vivax* and *P. falciparum* results to be quite out-of-place, given that the vast majority of the conclusions of the paper focused on *P. knowlesi*. I believe that the manuscript would benefit greatly from some restructuring of the results and placing greater emphasis on the fact that although the authors do not find evidence of sustained transmission of *P. knowlesi*, they do estimate non-zero R_c values for *P. knowlesi* which may indicate that there are stuttering chains of human-mosquito-human transmission. This is an important finding, and, although it must be considered in the context of the limitations of the method (i.e., no genetic data to actually prove human-mosquito-human transmission is occurring), it could warrant follow-up studies.

SPECIFIC COMMENTS

Introduction

Lines 93-99: Including results is strange and does not follow traditional scientific writing practice. Consider deleting.

Results:

Lines 104 – 106: This is unclear to me. Were all cases confirmed by molecular methods and 70% of those confirmations turned out to be *P. knowlesi*, or were 70% of cases, which were *P. knowlesi*, subject to confirmation by molecular methods? Reading the supplement, I am guessing that it is the latter, but this should be clearer in the main text.

Line 125: You mention 16,765 *P. knowlesi* cases screened, but on Line 105, you state that 23,143 *P. knowlesi* cases were subject to molecular methods. What happened to the 6,378 remaining cases? Did these turn out to not be *P. knowlesi*? If so, what were they?

Lines 136-137: Add a legend to Figure 3b. Additionally, it is not clear to me what the blue line signifies. Is this the logit fit from the supplementary material?

Line 148: Is the prior distribution from 0.001 to 1? If so, it's not really the prior distribution that is assuming that all cases are infected from human sources, but rather the value of the lower bound on the prior distribution. Consider rewording.

Line 148: Should "centring" be "centering"?

Line 149: Same recommendation as in Line 148. It's the value of the upper bound on the prior distribution on epsilon that assumes that all humans are infected from external zoonotic or imported sources rather than the prior distribution itself.

Lines 150-152: Please provide some measure of goodness of fit (e.g., AIC). The reader should be able to know how much better of a fit the value of 1 provides for epsilon.

Lines 154-157: I find the argument that $R_c < 1$ provides no evidence of sustained *P. knowlesi* transmission to be compelling. However, a non-zero R_c implies that some sub-critical transmission is occurring. This means that there are likely some stuttering chains of human-mosquito-human transmission. This is potentially an important finding, because it provides some initial evidence that there is some human-mosquito-human transmission of *P. knowlesi*. The authors should consider discussing this in greater detail. That said, I think it must be mentioned with the caveat that, due to the spatial and temporal proximity of cases, we may overestimate some local transmission.

DISCUSSION

General Comment: One thing that I find a bit out-of-place in the manuscript is the inclusion of the *P. falciparum* and *P. vivax* results. To me, it seems that the main focus (and the main conclusion) of the manuscript is that there is no evidence of sustained transmission of *P. knowlesi*. The results of the *P. vivax* and *P. falciparum* analysis seem to basically serve as a comparison that provides

credibility to the method. That is to say, the authors are basically using the *P. vivax* and *P. falciparum* analysis to show that, if there were sustained transmission of *P. knowlesi*, the authors would be able to detect it since there were able to detect sustained transmission of *P. falciparum* and *P. vivax*. If that's the case, I would recommend moving the *P. falciparum* and *P. vivax* analysis to the supplement. I do not think that the results shown in Figure 5, for instance, strengthen the paper.

In light of that, I would recommend replacing Figure 5 with further analysis of *P. knowlesi* transmission. In particular, the authors could include an estimate transmission network that shows that there may be some stuttering chains of *P. knowlesi* human-mosquito-human transmission. Additionally, it may be worth considering mapping where exactly the possible human-mosquito-human transmission is occurring. The authors could plot this alongside the likely spillover locations, given the estimated zoonotic cases of malaria. One way to do this is to model zoonotic spillover as a Poisson process as was done by Hugh Sturrock in this tutorial: https://github.com/disarm-platform/disarmr/blob/master/RMarkdowns/ppm_mgcv.md. I hope that the authors find this useful as I believe that this could help to streamline and strengthen the conclusions of their study.

Methods:

Lines 323-325: Would it be possible to have separate epsilons for zoonotic transmission and imported infections? In some sense, it seems that these are two different processes that are captured by the same epsilon?

Line 330: Is there evidence in the literature for choosing a value of 0.1 for delta? This study would benefit from either (1) estimating delta or (2) selecting multiple possible values of delta over a range (as was done for epsilon) and identifying the best values on the basis of AIC. In fact, it may be best to look over the possible finite combinations of delta and epsilon and choose the best fitting model on the basis of AIC if you are fixing this parameter.

Line 353: Which key parameters? Specify.

Supplementary Materials:

Table S1: This table should include more information on travel characteristics, number imported, indigenous, recurrences, etc.

Line 100: By modeling $\logit(\pi)$ as a linear regression, does this imply that the probability of infection is always increasing as a function of the days since symptom onset? If so, that functional form seems odd, because we know that at some point the probability of infection should decline (i.e., the overall shape should likely be unimodal). I think this is somewhat reflected in Fig. 3B where you do see a decline in proportion in later days.

General Comment: The study would benefit from a simulation study to demonstrate that the methodology is capable of estimating R_c given the characteristics of the dataset. The authors could simulate data that resembles the true data for a range of different R_c values and show that the inference framework is capable of recovering R_c .

General Comment: The study would additionally benefit from performing sensitivity analyses and model selection on the assumed value of epsilon and delta. The authors should plot how R_c changes given the value of epsilon and delta and report the AIC.

Reviewer #3 (Remarks to the Author):

Please see the attached PDF for comments.

“NO EVIDENCE OF SUSTAINED NONZOONOTIC PLASMODIUM KNOWLESI TRANSMISSION”

The present work concerns estimation of reproduction numbers (R_C) for different types of malaria in Malaysia; to this end, the authors use a spatially explicit statistical framework applied to a large dataset of infections from the Malaysian Ministry of Health. The authors illustrate that their methodology produces estimates for R_C for *P. Falciparum* and *P. Vivax* that are consistent with the success of malaria elimination programs in Malaysia. Moreover, they estimate R_C for *P. Knowlesi* and show that there is no evidence for sustained human-mosquito-human (i.e. nonzoonotic) transmission of *P. Knowlesi* in their framework.

The paper is well-written and the authors argue convincingly for their conclusion that *P. Knowlesi* transmission is almost exclusively down to zoonotic sources. The article is an interesting piece of work and is likely to be of interest to many in the field. I also think it is a valuable contribution to highlight the use of a statistical framework that explicitly accounts for the spatial extent of the infection processes. The paper could be improved by some modifications to the figures and I have some questions that would help to clarify aspects of the work, but I support publication of the paper with minor revisions.

Comments/questions:

- The authors state that “only local travel with the same village or subdistrict” was observed in the dataset and we know from Figure 1 that *P. Knowlesi* has significant spatial heterogeneity in its distribution. Could the authors comment on the justification for the (spatially) aggregated way that the data for *P. Knowlesi* is presented in Figure 4?
- Closely related to the last question, I found the spatio temporal estimates of R_C (Figure 5) among the most interesting results. I may have missed the explanation, but why did the authors not carry out this (or a similar spatio temporal R_C) analysis for *P. Knowlesi*? I understand that the corresponding probabilities are likely zero for *P. Knowlesi* but some similar plot of the spatial distribution of R_C for this species would be helpful.
- The y-axis in Figure 4 need to be adjusted for *P. Knowlesi* in order for us to be able to observe if there are temporal trends or not.
- How sensitive is the value of R_C to the value of δ ? Also, if movement is as highly localized as the authors suggest, does a model assuming that the villages are isolated patches produce essentially the same results or is the value of δ contributing to a better fit in the model?
- Is the human-to-mosquito ratio assumed constant in space throughout? I would expect there to be significant spatial heterogeneity in mosquito prevalence and so a spatially varying ratio may be at least as important to include as a time-varying mosquito-to-human ratio.

Minor comments/corrections:

- Labels/text are too small on Figures 4 and 5.
- L221, remove “a”.
- L325, mistake/typo with wording.
- Figure S1 is missing a colormap. Also, a colormap here might help to understand the impact of the value of δ (which I assume is 0.1 here).

REVIEWER COMMENTS

Reviewer #1 (Remarks to the Author):

The study explored the possibility of nonzoonotic transmission of *P.knowlesi* by fitting a mathematical model to cases reported in Malaysia. The model attempted to quantify the individual malaria case reproduction number R_c , which is the number of secondary human-mosquito-human cases from an index case, informing by three main quantities 1) the time of symptom onset, 2) spatial location of cases, and estimated serial interval (SI). The results shown that the R_c estimate of *P.knowlesi* is less than one, supporting that the feasibility of human-mosquito-human transmission is low. The authors also applied this same model to cases reported for *P.falciparum* and *P.vivax* and the results shown their R_c are larger than one in the past, confirming the historical human-to-human outbreaks of these malaria species. The findings of this manuscript is interesting, as nonzoonotic transmission of *P.knowlesi* was previously confirmed to be feasible in vivo but no evidence was found in epidemiological data. Defining the exact mode of transmission will help to optimize resources to prevent transmission of this malaria species. The methods used here was sound and interesting and could be used for other zoonosis diseases with available case report data. The manuscript was well written, clear, and ready to publish but still can benefit from some edits and suggestions.

Since the methods had a section focused on estimating SI, I think the introduction should briefly include some information about the natural progression of a *P.knowlesi* case, like incubation period, infectious period, recovery rate, proportion of mortality or disability. This information was scattered around the manuscript but without a systematic summary beforehand, it may confuse people who is not familiar with the disease.

We agree these details are important to clarify and have added more background in the introduction (Lines 62-64).

Related to the calculation of SI, I find it confusing that the proportion of infectious increases with the days since symptom onset. I wonder if this is the natural progression of the disease since I expect the proportion of infectious will eventually go down as people recover. The supplement information mentioned that the proportion infectious was kept to be the same after day 14, this information should be mentioned in the main text.

The increase in infectiousness (and failure to see a decrease in infectiousness) is likely due to the very short time between symptoms and treatment in the majority of cases reported (Lines 290-291). We have clarified this within the text but the overall probability distribution used for the SI incorporates the range of plausible SI distributions.

One of the critics in the model when I was reading the main text is that the geographic variation of Malaysia was not accounted for when deriving the SI, but I did eventually see this point made in the supplement information. I think it is worth to bring this issue up to the main text and discuss it as a limitation of the study.

We agree this is a limitation and have added this in the conclusions as a priority for future studies (Lines 226 – 228).

Lines 240-244: this point could be clearer if the authors articulate how the results going to change if they did not account for the asymptomatic or unreported infections. The authors should also include some information about the competency of asymptomatic transmission.

No data is available on the competency of asymptomatic infections; however, most asymptomatic infections detected in people have been low densities (submicroscopic) and

unlikely to have high densities of gametocytes.

Although the authors have documented the details of their methods in the supplement information, codes should still be required for the ease of reproducing the results.

While the data cannot be shared due to ethics restrictions, we have referenced the code repository where the model code can be accessed and the paper described the model structure and validation (Lines 341 – 342).

Reviewer #2 (Remarks to the Author):

ASSESSMENT

In this study, Fornace et al. leveraged high-resolution data from Malaysia to address whether there is sustained human-mosquito-human transmission of *Plasmodium knowlesi*. The authors use a well-established transmission network approach to estimate R_c , the basic reproduction number under control. They conclude that, because the estimated R_c falls below one and lies very close to zero, there is no evidence of sustained human-mosquito-human transmission of *P. knowlesi*. I really enjoyed this study and find the conclusions compelling. I mainly just have clarification and recommendations that I think will strengthen the paper when it is published in its final form. In particular, I believe that this study would benefit from a simulation study to prove that the inference framework is capable of estimate the correct values of R_c as well as sensitivity analyses to evaluate the sensitivity of the conclusions to the choice of ϵ and δ .

We agree this is an important component of model development and validation. This study was informed by simulation studies conducted by Routledge et. al, 2021 and we have highlighted this within the text (Lines 341 – 342, including link to GitHub repository). This study demonstrates the effect of specific priors or assumptions around ϵ and δ across different transmission settings.

Finally, I found the inclusion of the *P. vivax* and *P. falciparum* results to be quite out-of-place, given that the vast majority of the conclusions of the paper focused on *P. knowlesi*. I believe that the manuscript would benefit greatly from some restructuring of the results and placing greater emphasis on the fact that although the authors do not find evidence of sustained transmission of *P. knowlesi*, they do estimate non-zero R_c values for *P. knowlesi* which may indicate that there are stuttering chains of human-mosquito-human transmission. This is an important finding, and, although it must be considered in the context of the limitations of the method (i.e., no genetic data to actually prove human-mosquito-human transmission is occurring), it could warrant follow-up studies.

We can appreciate this point but would interpret these results with caution as there is no clear evidence of any sustained *P. knowlesi* transmission or clearly linked clusters of cases. We have included additional maps of *P. knowlesi* R_c estimates within the Supplementary Information to enable visualisation of areas with the highest probabilities of nonzoonotic transmission. We have included an additional point within the discussion that these R_c estimates may be useful to guide future studies on stuttering chains of transmission (Lines 228-232). We additionally feel the contrast between *P. knowlesi* and human malaria species is key to demonstrate the differences in transmission between species (see response to discussion below).

SPECIFIC COMMENTS

Introduction

Lines 93-99: Including results is strange and does not follow traditional scientific writing practice. Consider deleting.

We have structured this article similarly to other research published by Nature Communications and the stated guidelines to include the major findings and results within the last paragraph of the introduction. While we believe this aids interpretation of the manuscript, we will defer to any editorial recommendations if requested to change this.

Results:

Lines 104 – 106: This is unclear to me. Were all cases confirmed by molecular methods and 70% of those confirmations turned out to be *P. knowlesi*, or were 70% of cases, which were *P. knowlesi*, subject to confirmation by molecular methods? Reading the supplement, I am guessing that it is the latter, but this should be clearer in the main text.

We have reworded this sentence for clarity. Yes, all suspected *P. knowlesi* cases diagnosed by microscopy were confirmed by molecular methods (Lines 107-109).

Line 125: You mention 16,765 *P. knowlesi* cases screened, but on Line 105, you state that 23,143 *P. knowlesi* cases were subject to molecular methods. What happened to the 6,378 remaining cases? Did these turn out to not be *P. knowlesi*? If so, what were they?

This refers to the subset of *P. knowlesi* cases which were examined for gametocytes. We have clarified this within the text (Line 123).

Lines 136-137: Add a legend to Figure 3b. Additionally, it is not clear to me what the blue line signifies. Is this the logit fit from the supplementary material?

Yes, the line represents the logit fit and we have clarified this within the figure legend.

Line 148: Is the prior distribution from 0.001 to 1? If so, it's not really the prior distribution that is assuming that all cases are infected from human sources, but rather the value of the lower bound on the prior distribution. Consider rewording.

We have clarified that these values refer to the range of mean prior values assessed (Line 139-140). These were different priors used for epsilon with the means centring on different values rather than a bound on the upper and lower limits.

Line 148: Should "centring" be "centering?"

This is a difference between British and American spelling but we are happy to follow any journal recommendations to standardise formatting.

Line 149: Same recommendation as in Line 148. It's the value of the upper bound on the prior distribution on epsilon that assumes that all humans are infected from external zoonotic or imported sources rather than the prior distribution itself.

We have clarified this above and within this sentence (Line 142). We have additionally included a supplementary table (S4) including the mean and SD values used for priors in the sensitivity analysis.

Lines 150-152: Please provide some measure of goodness of fit (e.g., AIC). The reader should be able to know how much better of a fit the value of 1 provides for epsilon.

We have included an additional table on model selection statistics in the Supplementary Information (Table S4).

Lines 154-157: I find the argument that $R_c < 1$ provides no evidence of sustained *P. knowlesi* transmission to be compelling. However, a non-zero R_c implies that some sub-critical transmission is occurring. This means that there are likely some stuttering chains of human-mosquito-human transmission. This is potentially an important finding, because it provides some initial evidence that there is some human-mosquito-human transmission of *P. knowlesi*. The authors should consider discussing this in greater detail. That said, I think it must be mentioned with the caveat that, due to the spatial and temporal proximity of cases, we may overestimate some local transmission.

We have included additional text about the need to assess potential stuttering chains of transmission (Lines 229 – 230) but we would hesitate to conclude this is occurring. Please see comment below for further details.

DISCUSSION

General Comment: One thing that I find a bit out-of-place in the manuscript is the inclusion of the *P. falciparum* and *P. vivax* results. To me, it seems that the main focus (and the main conclusion) of the manuscript is that there is no evidence of sustained transmission of *P. knowlesi*. The results of the *P. vivax* and *P. falciparum* analysis seem to basically serve as a comparison that provides credibility to the method. That is to say, the authors are basically using the *P. vivax* and *P. falciparum* analysis to show that, if there were sustained transmission of *P. knowlesi*, the authors would be able to detect it since there were able to detect sustained transmission of *P. falciparum* and *P. vivax*. If that's the case, I would recommend moving the *P. falciparum* and *P. vivax* analysis to the supplement. I do not think that the results shown in Figure 5, for instance, strengthen the paper.

We feel strongly that the results from nonzoonotic malaria species strengthen the conclusions around the differences in transmission between *P. knowlesi* and other species. Within many areas of Malaysia, the mosquito vectors are the same for all malaria species. As the treatment seeking behaviour and movement patterns of the human population are the same, this clearly demonstrates these parameters can identify potential nonzoonotic chains of transmission. As other reviewers highlighted the utility of these figures, we feel these should remain in the main text but have also updated the *P. knowlesi* maps to show higher resolution distributions of case numbers reported.

In light of that, I would recommend replacing Figure 5 with further analysis of *P. knowlesi* transmission. In particular, the authors could include an estimate transmission network that shows that there may be some stuttering chains of *P. knowlesi* human-mosquito-human transmission. Additionally, it may be worth considering mapping where exactly the possible human-mosquito-human transmission is occurring. The authors could plot this alongside the likely spillover locations, given the estimated zoonotic cases of malaria. One way to do this is to model zoonotic spillover as a Poisson process as was done by Hugh Sturrock in this tutorial: https://github.com/disarm-platform/disarmr/blob/master/RMarkdowns/ppm_mgcv.md. I hope that the authors find this useful as I believe that this could help to streamline and strengthen the conclusions of their study.

While we can appreciate the value of approaches to model the potential distribution of spillover locations, we feel these methods are outside the focus of this paper and would benefit from a separate and extensive piece of research modelling key habitat types and distribution of wildlife reservoirs. Although R_c estimates were above zero for some *P. knowlesi* cases, no estimates exceeded 1. The R_c value is calculated as the sum of the normalised likelihood of the case infecting other cases and an individual R_c value less than 1 indicates a low likelihood that this case was the source for any other infections. Because of this, there were no clear chains of transmission which could be identified. This also limited any potential mapping due to excess zeroes. However, we have included maps of the mean and maximum point estimates of R_c within the supplementary information. We have additionally included references to these figures in the caption for Figure 5 to explain why similar maps were not created for *P. knowlesi*.

Methods:

Lines 323-325: Would it be possible to have separate epsilons for zoonotic transmission and imported infections? In some sense, it seems that these are two different processes that are captured by the same epsilon?

Within this model, epsilon refers to the probability of a case being infected by a source outside the surveillance dataset (e.g. either zoonotic sources or nonreported cases). Within this model, imported cases (observed by the surveillance dataset) are assessed for possible onward transmission but are assumed not to be infected from other cases within the surveillance dataset. Due to this, epsilon does not capture the probability of importation as importation status is recorded within the surveillance dataset (Lines 302 – 308).

Line 330: Is there evidence in the literature for choosing a value of 0.1 for delta? This study would benefit from either (1) estimating delta or (2) selecting multiple possible values of delta over a range (as was done for epsilon) and identifying the best values on the basis of AIC. In fact, it may be best to look over the possible finite combinations of delta and epsilon and choose the best fitting model on the basis of AIC if you are fixing this parameter.

We chose to fit this parameter based on the travel history collected within the Malaysian surveillance dataset, the high proportion of cases diagnosed within the same district and the criteria used for classification of imported cases. This supported the assumption that most travel was local (within the same district) and the probability of longer range travel was low. This corresponds to the probability distribution for distance travelled included in Figure S1.

Line 353: Which key parameters? Specify.

We have specified the parameters here (Lines 337).

Supplementary Materials:

Table S1: This table should include more information on travel characteristics, number imported, indigenous, recurrences, etc.

We have included an additional table with a breakdown of key demographic characteristics by reported parasite species in Table S2.

Line 100: By modeling $\text{logit}(\pi)$ as a linear regression, does this imply that the probability of infection is always increasing as a function of the days since symptom onset? If so, that functional form seems odd, because we know that at some point the probability of infection

should decline (i.e., the overall shape should likely be unimodal). I think this is somewhat reflected in Fig. 3B where you do see a decline in proportion in later days.

We agree that this should be unimodal after recovery but the short time between symptom onset and treatment meant there were insufficient cases untreated for a long enough proportion of time to assess the impacts of recovery (see response to reviewer 1). Rather than use a fixed value, we have used a distribution to include a range of plausible serial interval distributions.

General Comment: The study would benefit from a simulation study to demonstrate that the methodology is capable of estimating R_C given the characteristics of the dataset. The authors could simulate data that resembles the true data for a range of different R_C values and show that the inference framework is capable of recovering R_C .

This study was informed by extensive sensitivity analyses conducted as part of a simulation study conducted by Routledge et. al. This has been highlighted within the text (Lines 344-345).

General Comment: The study would additionally benefit from performing sensitivity analyses and model selection on the assumed value of epsilon and delta. The authors should plot how R_C changes given the value of epsilon and delta and report the AIC.

We have included a more detailed breakdown of the AICc values for the sensitivity analysis on epsilon in the SI (Table S4 and S5). Within this table, we have additionally included summary statistics on the mean R_C value estimated and, for *P. falciparum* and *P. vivax*, the proportion of R_C estimates over 1.

We have not conducted a similar sensitivity analysis for delta as this parameter is based on the reported travel history and location of residence and diagnosis within the surveillance dataset. Additionally, within this modelling framework, any cases classified as imported cannot be a secondary case resulting from another case within this surveillance data. Within Figure S1, this shows the probability of 2 cases being linked based on Euclidean distance; this corresponds to the majority of travel occurring within the same district. We have additionally included a breakdown of cases by species and classification as indigenous within Table S2.

Reviewer #3 (Remarks to the Author):

The present work concerns estimation of reproduction numbers (R_C) for different types of malaria in Malaysia; to this end, the authors use a spatially explicit statistical framework applied to a large dataset of infections from the Malaysian Ministry of Health. The authors illustrate that their methodology produces estimates for R_C for *P. falciparum* and *P. vivax* that are consistent with the success of malaria elimination programs in Malaysia. Moreover, they estimate R_C for *P. knowlesi* and show that there is no evidence for sustained human-mosquito-human (i.e. nonzoonotic) transmission of *P. knowlesi* in their framework. The paper is well-written and the authors argue convincingly for their conclusion that *P. knowlesi* transmission is almost exclusively down to zoonotic sources. The article is an interesting piece of work and is likely to be of interest to many in the field. I also think it is a valuable contribution to highlight the use of a statistical framework that explicitly accounts for the spatial extent of the infection processes. The paper could be improved by some modifications to the figures and I have some questions that would help to clarify aspects of the work, but I support publication of the paper with minor revisions.

Comments/questions:

The authors state that "only local travel with the same village or subdistrict" was observed in the dataset and we know from Figure 1 that *P. Knowlesi* has significant spatial heterogeneity in its distribution. Could the authors comment on the justification for the (spatially) aggregated way that the data for *P. Knowlesi* is presented in Figure 4?

We did not fit geostatistical models for *P. knowlesi* as all R_C estimates were below 1. We aggregated these data by district to better display widescale spatial trends in reported case numbers but we can appreciate more detail would be useful. To address this point, we have included mean and maximum estimates of R_C for *P. knowlesi* by village in the SI. We have highlighted this limitation and the additional figures in the caption for Figure 5.

Closely related to the last question, I found the spatio temporal estimates of RC (Figure 5) among the most interesting results. I may have missed the explanation, but why did the authors not carry out this (or a similar spatio temporal RC) analysis for *P. Knowlesi*? I understand that the corresponding probabilities are likely zero for *P. Knowlesi* but some similar plot of the spatial distribution of RC for this species would be helpful.

We can appreciate this point and have included additional maps of mean and maximum R_C estimates for *P. knowlesi* per year in the Supplementary Information to enable visualisation of priority areas for future surveillance. We haven't fit similar geostatistical models of these estimates as all estimates are less than 1 and there would be a 0% probability of an R_C exceeding 1. We have clarified this within the text (caption Figure 5) and included supplementary figures.

The y-axis in Figure 4 need to be adjusted for *P. Knowlesi* in order for us to be able to observe if there are temporal trends or not.

We have rescaled this figure to better show temporal trends of *P. knowlesi*. If further corrections are requested, we can adjust these plots.

How sensitive is the value of RC to the value of δ ? Also, if movement is as highly localized as the authors suggest, does a model assuming that the villages are isolated patches produce essentially the same results or is the value of δ contributing to a better fit in the model?

We have included additional details on a sensitivity analysis exploring the effects of varying delta (Routledge et. al, 2021) with the accompanying code repository. Within this surveillance dataset, locations of recent travel and possible infection were recorded as GPS points and/or village names from detailed case investigations. The distribution of these points informed the value for delta. We have additionally included the breakdown of indigenous cases by species based on the classification as imported (Table S2).

Is the human-to-mosquito ratio assumed constant in space throughout? I would expect there to be significant spatial heterogeneity in mosquito prevalence and so a spatially varying ratio may be at least as important to include as a time-varying mosquito-to-human ratio.

We agree with the reviewer that there is likely significant spatial heterogeneity in the human-mosquito biting rate. Within this model, we used a distribution of possible serial intervals rather a fixed to encompass the range of probabilities across this region and to account for

uncertainty in this estimate. As these parameters have identified chains of nonzoonotic malaria transmitted by the same mosquito vectors, we feel this parameter range is able to identify linked cases. However, we also included this as an area for further research in the discussion (Lines 231-233).

Minor comments/corrections:

_ Labels/text are too small on Figures 4 and 5.

We can appreciate this point as figures were initially submitted within a word document for the initial submission. We have uploaded higher resolution figures and can alter as needed.

_ L221, remove \a".

_ L325, mistake/typo with wording.

We have edited the manuscript and corrected typos.

_ Figure S1 is missing a colormap. Also, a colormap here might help to understand the impact of the value of δ (which I assume is 0:1 here).

We have added a legend to this figure.

REVIEWERS' COMMENTS

Reviewer #2 (Remarks to the Author):

I appreciate the time and effort made that the authors took to address the comments and concerns raised by myself and the other reviewers.

My only last remark is further clarification in the text on the choice of delta of 0.1. The authors state that in the response that this was fit based on the surveillance data, but in the main text it said it was chosen based on a transmission distance of 10km. Is this a rough approximation of the distance of the villages in the area? I am not familiar with the location geography so I will defer to the authors. Alternatively, can the authors provide some evidence in the literature to cite as support for this value. If not, then I would appreciate a sensitivity analysis looking at how the results change with the assumed value of this parameter.

Reviewer #3 (Remarks to the Author):

I am satisfied with the answers from the authors in relation to my comments on the first version of the paper and they have addressed all of the concerns raised. I am happy to recommend this paper for publication.

REVIEWER COMMENTS

Reviewer #2 (Remarks to the Author):

I appreciate the time and effort made that the authors took to address the comments and concerns raised by myself and the other reviewers.

My only last remark is further clarification in the text on the choice of delta of 0.1. The authors state that in the response that this was fit based on the surveillance data, but in the main text it said it was chosen based on a transmission distance of 10km. Is this a rough approximation of the distance of the villages in the area? I am not familiar with the location geography so I will defer to the authors. Alternatively, can the authors provide some evidence in the literature to cite as support for this value. If not, then I would appreciate a sensitivity analysis looking at how the results change with the assumed value of this parameter.

We can appreciate this comment from the reviewer and have added further clarification on why we chose this value. This includes Line 318 in the main text and the following more detailed explanation within the SI:

“The fixed value for the spatial parameter corresponded to most cases being infected within a 10km radius of their reported residence location; this parameter was obtained based on reported travel history from case investigations reporting most individuals remaining within the same village or district prior to their diagnosis. Individuals with a history of long-range travel were classified as imported cases according to Malaysian Ministry of Health surveillance guidelines.”

We feel the value of delta is justified by the reported travel history, sizes of districts and villages within Malaysia and the classification of long-range travel as imported. Our previous studies of movement and disease exposure (e.g. Fornace et. al, 2019, *Elife*) additionally support local transmission. However, we have also included a link to the simulation study and all code showing the impacts of varying delta parameters on estimates of R_C (Lines 343 – 344).

Reviewer #3 (Remarks to the Author):

I am satisfied with the answers from the authors in relation to my comments on the first version of the paper and they have addressed all of the concerns raised. I am happy to recommend this paper for publication.